# Clinical Role of Serum miR107 in Type 2 Diabetes and Related Risk Factors

**DOI:** 10.3390/biom12040558

**Published:** 2022-04-08

**Authors:** Diana Šimonienė, Darius Stukas, Albertas Daukša, Džilda Veličkienė

**Affiliations:** 1Department of Endocrinology, Lithuanian University of Health Sciences (LUHS), 50161 Kaunas, Lithuania; dzilda.velickiene@lsmuni.lt; 2Laboratory of Surgical Gastroenterology, Institute for Digestive Research, Lithuanian University of Health Sciences (LUHS), 44307 Kaunas, Lithuania; darius.stukas@lsmuni.lt (D.S.); albertas.dauksa@lsmuni.lt (A.D.); 3Department of Surgery, Lithuanian University of Health Sciences (LUHS), 50161 Kaunas, Lithuania; 4Institute of Endocrinology, Lithuanian University of Health Sciences (LUHS), 44307 Kaunas, Lithuania

**Keywords:** insulin resistance 1, type 2 diabetes 2, miR-107

## Abstract

Background: As the diagnostic and treatment options for diabetes improve, more attention nowadays is being paid to the exact identification of the etiopathological mechanism of type 2 diabetes (T2DM). Insulin resistance (IR) is a pathogenetic background for T2DM. Several studies demonstrate that miRNAs play an important role in systemic inflammation and thus in T2DM pathogenesis. Overexpression of miR-107 may cause an imbalance of glucose homeostasis, obesity, and dyslipidemia, by regulating insulin sensitivity through the insulin signaling pathway. Methods: 53 patients with T2DM and 54 nondiabetic patients were involved in the study. This study aimed to examine whether miR-107 expression in the serum of patients with diabetes was different from the control group (non-diabetic) and whether miR-107 expression correlated with lipid levels, BMI, and other factors, and finally, with insulin resistance in general. Results: miR-107 expression was higher in the T2DM group than in the control group (1.33 versus 0.63 (*p* = 0.016). In general, miR-107 expression was directly and positively associated with BMI (*r* = 0.3, *p* = 0.01), age (*r* = 0.3, *p* = 0.004), and male gender (*p* = 0.006). Moreover, miR-107 was related to dyslipidemia: Patients with higher miR-107 levels had lower HDL levels (in the control group: *r* = −0.262, *p* = 0.022 vs. diabetic group: *r* = −0.315, *p* = 0.007). Finally, the overexpression of miR-107 was associated with higher HOMA-IR in the diabetic group (*r* = 0.373, *p* = 0.035). Conclusion: MiR-107 expression is higher among diabetic patients than that of nondiabetic control subjects. Higher miR-107 levels are also related to dyslipidemia (lower HDL levels)—in the general cohort and non-diabetic subjects. Moreover, higher miR-107 expression is related to insulin resistance in the diabetic group. In general, higher miR-107 expression levels are related to a higher BMI, older age, and the male gender.

## 1. Introduction

Type 2 diabetes mellitus (T2DM) is a chronic disorder triggered by an imbalanced metabolism [1] and is one of the most prevalent diseases in the world. Many studies have demonstrated the natural history of insulin resistance (IR) and β-cell insulin secretion in T2DM. Moreover, a central role of IR in the development of T2DM and other cardiometabolic disturbances (dyslipidemia, adiposity, and high blood pressure) has been suggested [2]. The precise pathophysiological mechanism of IR is still largely discussed. Recent studies have contributed to a deeper understanding of the underlying molecular mechanisms. A series of studies have provided evidence of a genetic mechanism linked to IR [3,4]. Nowadays, several studies demonstrate that microRNAs (miRNAs, miRs) play an important role in the etiology and pathogenesis of T2DM and obesity [5,6], through IR regulation pathways [7].

MiRNAs participate in the regulation of protein-encoding genes processes. They are single-stranded, non-coding RNA molecules, approximately 22-nucleotides long, which play a crucial role in the regulation of gene expression, by binding to the RNAs to destabilize them or inhibit their translation [8]. It has been demonstrated that any single miRNA can be directly responsible for the regulation of many proteins and that any single protein-coding gene can be affected by more than one miRNA [9]. It is known that the human genome encodes more than 2000 miRNAs. Approximately 70 miRNAs showed elevated levels and around 100 miRNAs showed reduced levels in blood samples of T2D patients [10]. At least 25 miRNAs are primarily involved in insulin resistance (e.g., miR-15b, -195, -320, -223, -378, -29a, 103/107, etc.) [11]. In the last decade, researchers have discovered that miRNAs exist in serum and plasma. Some authors suggest that serum and plasma miRNAs may serve as valuable biomarkers of disease [12]. Adipose tissue can release miRNAs that can act as signaling molecules [13].

MiR-107 is one of the most promising miRNAs in evaluating diabetes and insulin sensitivity. MiR-107 has been previously found to be a critical miRNA in glucose metabolism and insulin sensitivity [14]. Experimental studies have also demonstrated that the overexpression of MiR-107 may cause an imbalance of glucose homeostasis, in relation to insulin sensitivity [15]. Moreover, experiments have demonstrated a correlation between the expression of miR-103/107 and adipose tissue mass by accelerating its adipogenesis. Additionally, an implication in lipid metabolism was also demonstrated. According to the authors, these two miRNAs target genes that are involved in the regulation of cellular acetyl-CoA and lipid levels [16]. Finally, miR-107 was found to inhibit cyclin-dependent kinase 6 (CDK6) expression in adipocytes, which regulates adipogenesis and lipid storage [17], which leads to obesity and dyslipidemia. According to Foley et al. [18], miR-107 may be a key miRNA, linking inflammatory processes in T2DM, IR, and obesity. Trajkovsky et al. identified caveolin-1, a critical regulator of the insulin receptor, as a direct target gene of miR-107 [14]. We hypothesize that miR-107 could act as a messenger molecule that is highly important in the orchestration of the inflammatory process and its clinical expression, namely obesity, insulin resistance, diabetes, and dyslipidemia.

This study aimed to examine whether miR-107 expression in the serum of patients with diabetes was different from that of the control group (non-diabetic) and whether miR-107 expression correlated with lipids level, BMI, and other factors, in comparison with insulin resistance in general.

## 2. Materials and Methods

### 2.1. Patient Selection

Fifty-three patients with T2DM and 54 non-diabetic outpatients, aged over 18 years old, were included in the study. This study was conducted at the tertiary care level of the hospital of the Lithuanian University of Health Sciences, over a period of 2 years (2020–2021).

### 2.2. Study Design

Case-control study. Patients with T2DM were assigned to the case group. Participants that had not been diagnosed with diabetes or prediabetes were assigned to the control group. Each participant of the control group was matched with a case group subject, according to gender and age.

The study was approved by the Kaunas Regional Biomedical Research Ethics Committee, (P1-BE-2-29/2017, No SRI—02 version 5, 16 July 2021), before the start of the study.

### 2.3. Inclusion and Exclusion Criteria

Inclusion criteria for diabetic (case) group: Diagnosed T2DM, aged over 18 years old, treatment with insulin, with or without metformin.

Inclusion criteria for non-diabetic (control) group: No diabetes or prediabetes, matched with case group according to age and gender.

Exclusion criteria: Advanced kidney disease or end-stage kidney impairment (GFR < 30 mL/min), active cardiovascular disease (for example acute myocardial infarction or unstable angina), or any other vascular disease, oncological disease within the six months before the start of the investigation, ongoing treatment with glucocorticoids.

#### Serum Samples

Overall, 108 serum samples from different participants were analyzed. Fifty-three serum samples were obtained from insulin-treated patients with T2DM, and 54 samples were obtained from the control subjects.

### 2.4. Methods

#### 2.4.1. Measurements, Laboratory Tests

Weight and height were measured while participants were wearing light clothing and no shoes. Height was assessed with 0.1 cm accuracy using a wall stadiometer. Weight was measured with a digital scale with 0.1 kg accuracy. According to the results of weight and height, the body mass index (BMI) was calculated. Obesity was defined as a body mass index (BMI) above ≥30 kg/m^2^. Overweight was defined as a BMI above ≥27 kg/m^2^. Age was taken from the electronic medical records. On the day of the consultation with an endocrinologist, fasting insulinemia and fasting glycemia were measured in a hospital laboratory. The Insulin Resistance Index (HOMA-IR) was calculated using the mathematical equation (insulin (mU/mL)*glucose (mmol/L)/22.5), and the ratio of HOMA-IR was used as a measure of insulin resistance. Insulin resistance was defined as a HOMA-IR above ≥2.5, according to literature recommendations [19]. For miR-107 detection, 5 mL of the serum sample was also taken. The blood was drawn, and the serum was separated by double centrifugation. Fasting glycemia and HbA1c values for control group subjects were also analyzed. In cases where fasting glycemia was above 6 mmol/L or HbA1c was above 5.7%, a 2 h oral glucose tolerance test (OGTT) was performed on the control group participants. Subjects with diagnosed T2DM but with abnormal test results and patients with prediabetes were excluded from the study.

We also evaluated the level of lipids at the fasting state. The analysis was performed on the day of the visit to the endocrinologist, at the same time as the rest of the tests. According to the National Cholesterol Education Program (NCEP), Adult Treatment Panel III (ATP III), dyslipidemia is defined as plasma triglycerides >1.7 mmol/L, and high-density lipids (HDL) cholesterol <1.03 mmol/L in men or <1.29 mmol/L in women. [20].

##### 2.4.2. microRNA Isolation, Reverse Transcription, and Quantitative PCR

A miRNAeasy serum/plasma Kit (Qiagen, Hilden, Germany) was used to isolate the desired microRNA from the serum samples, according to the manufacturer’s protocol. Without any delay, the isolated microRNA was converted to cDNA using a TaqMan MicroRNA reverse transcription kit (Applied Biosystems, Vilnius, Lithuania) and TaqMan MicroRNA assays (Applied Biosystems, Pleasanton, CA, USA) with a microRNA-107 primer and an RNU6B primer, to serve as reference microRNA. The converted samples were stored at −20 °C, pending further analysis.

The expression levels of microRNAs were measured using a quantitative reverse transcriptase-polymerase chain reaction (qRT-PCR). Amplification was performed using TaqMan miRNA assays (Applied Biosystems, Pleasanton, CA, USA) and TaqMan Universal Master Mix II (Applied Biosystems, Vilnius, Lithuania) on a 7500 fast real-time PCR system (Life Technologies, Carlsbad, CA, USA), also according to the manufacturer’s protocol. The expression levels of microRNA-107 were normalized to RNU6B and expression changes were calculated using the 2^−∆Ct^ method.

### 2.5. Statistical Methods

Statistical analyses were performed using the Statistical Package for Social Sciences (SPSS), Version 27. Descriptive statistics were used to summarize all measurements. The Kolmogorov–Smirnov test allowed us to check that the results of the samples corresponded to a normal or abnormal distribution. In cases of normal distribution, the main descriptive data were expressed as mean ± SD, and comparisons between two means of independent samples were analyzed using Student’s t-test. In cases of abnormal distribution, the main descriptive data were expressed as medians (min–max). Comparisons between two means of independent nonparametric samples were analyzed using the Whitney–Mann test. To compare categorical variables, the chi-squared test χ2 test was performed. Quantitative data were tested with Pearson‘s (in cases of normal distribution) or Spearman’s (in cases of abnormal distribution) correlation coefficients (r). A multivariate logistic regression method was used to determine the most important relationship factors. Odds ratios (OR) and 95% confidence intervals (CI) for dyslipidemia in diabetes were calculated.

The level of statistical significance was set as *p* < 0.05.

## 3. Results

The main descriptive data (such as age, gender, BMI, lipids, expression of miR-107, and other factors) of the T2DM and control groups are presented in Table 1.

Study participants were matched according to age and gender, so both groups were similar (the mean age was 64 years, and the male/female ratio was approximately 50/50). However, we found a significant difference between weight in both groups: Many subjects of the T2DM group met the criteria for obesity, whilst many subjects of the control group were classified as overweight. The average HbA1c level of the diabetic group was 8.23%, demonstrating that subjects with differently adjusted diabetes were included in the study. The mean HbA1c level of the control group was 5.46%, demonstrating that the participants’ mean HbA1c did not reach the levels of prediabetes, the same as normal fasting glycemia.

Total cholesterol and low-density lipid levels (LDL) were lower in the diabetic group because of the use of statins (used by 74% of diabetic patients). However, according to the NCEP ATP III definition, real dyslipidemia is evaluated with triglycerides and HDL levels, which were statistically and significantly different across the groups. According to the NCEP ATP III definition, the rate of dyslipidemic patients was higher in the diabetes group than in the control group (76.9% versus 36.7%, *p* < 0.001), despite the use of statins.

HOMA-IR was used as a method to evaluate IR, to compare the expression of miR-107 between groups: In the control group, the mean HOMA-IR was in a normal range, while in the diabetic group, the mean HOMA-IR was significantly higher (1.9 versus 6.5 (*p* = 0.000)). Overall, it was found that miR-107 expression was higher in the T2DM group than in the control group (1.33 versus 0.63 (*p* = 0.016) (see Figure 1). MiR-107 expression did not differ in metformin users compared to nonusers in the T2DM group (1.25 [0.427–2.3] vs. 1.556 [0.611–3.269] (*p* = 0.522).

For a deeper analysis of dyslipidemia risk factors, BMI, HOMA-IR, and miR-107 levels were evaluated in relation to lipid levels. BMI correlated with the total cholesterol level *r* = −2.37 (*p* = 0.041), HDL level *r* = −0.493, *p* < 0.001, and with triglycerides *r* = 0.365, *p* = 0.002. HOMA-IR correlated with the total cholesterol level *r* = −0.401, *p* = 0.009, LDL level *r* = −0.331, *p* = 0.032, and HDL *r* = −0.428, *p* = 0.005. However, triglycerides did not demonstrate a relation with HOMA-IR.

Meanwhile, in the general cohort, miR-107 correlated negatively with the HDL levels (*r* = −0.315, *p* = 0.007) (see Figure 2), whereas in the control group, the negative correlation was not so significant *r* = −0.262, *p* = 0.022 (see Figure 3). This shows that a higher miR-107 expression was found among those with a lower HDL level. HDL did not show any association with miR-107 in the diabetic group.

MiR-107 was detected in 53 diabetic patients’ serum and matched non-diabetic patients serum by qRT-PCR and the miR-107 abundance was normalized to RNU6B.

According to our study aims, we wanted to analyze other factors that regulated the levels of miR-107 in general and in the test groups. We analyzed the relation of various factors (BMI, age, and gender) with miR-107 expression. It was found that miR-107 expression was directly and positively associated with BMI (*r* = 0.3, *p* = 0.01) (Figure 4) and age (*r* = 0.3, *p* = 0.004) (Figure 5) in the whole sample. While BMI did not demonstrate a significant association with miR-107 in the separate groups, we speculate that it is due to the small number of participants in each group.

Age in the diabetic group did not demonstrate a significant relation with miR-107, unlike the control group, where miR-107 expression was directly and positively associated with age (*r* = 0.38, *p* = 0.02), especially in women (*r* = 0.402, *p* = 0.017).

Moreover, the overexpression of miR-107 was generally noticed in men (*p* = 0.006) (Figure 6).

In the control group, miR-107 expression was significantly different (*p* = 0.001 based on the Mann-Whitney test) between genders (median [25–75%]): 1.72 [0.65–4.06] for men and 0.46 [0.18–1.35] for women. In the diabetic cohort, a correlation between higher miR-107 expression and male gender (*r* = 0.395, *p* = 0.05) was found. In general, the male gender is related to higher miR-107 expression (*p* = 0.002).

There was no correlation between miR-107 expression and obesity/overweight with an increased HOMA-IR.

Finally, the correlation of miR-107 expression with HOMA- IR was evaluated. Only in the diabetic group did miR-107 expression correlate with the HOMA-IR level (*r* = 0.373, *p* = 0.035). This means that the expression of miR-107 was highest in patients with the highest HOMA-IR (or at most insulin resistant cases), but only in the diabetic group.

## 4. Discussion

MiR-107 is well studied in adipocytes in an insulin-resistance model with rodents [19]. With pathological conditions, such as T2DM, there are only a few in vivo tests to prove the role of miR-107 in humans. This is the first study that evaluates the expression of miR-107 in relation to diabetes, insulin resistance, lipids, and BMI.

Our study showed that obese diabetic patients had increased miR-107 expression compared to the non-diabetic overweight group. Thus, weight relates to miR-107 expression level. In the current study, it was demonstrated that miR-107 was positively associated with BMI, and patients with a higher BMI had higher miR-107 levels. The same results were shown in a Turkish study, where authors evaluated miR-107 levels in plasma in obese children [21]. They postulated that obesity increased miR-107 levels in plasma in children. Trajkovski et al. postulated that miR-107 inhibits adipocyte differentiation and that treatment with anti-miRs could improve insulin sensitivity by increasing adipocyte differentiation [14]. Many experiments have now shown a correlation between miR-107 expression and adipose tissue [16].

The present results also showed a significant positive correlation between miR-107 and HOMA-IR, but only in the diabetic group. These findings were consistent with Qian et al., who reported that miR-103 and miR-107 may be potential molecular markers for insulin resistance [22], and also with Tuzlukaya et al. [21] who demonstrated a positive correlation between miR-107 and HOMA-IR, used to evaluate insulin resistance. The present study compared non-diabetic overweight subjects with obese diabetic patients with advanced diabetes (median diabetes duration was 15 years). Our study demonstrated that in advanced diabetes, the median miR-107 expression was approximately 1.33. Meanwhile, in the overweight control group, the median expression of miR-107 was 0.63 We noticed that the expression of miR-107 is twice as low in the absence of diabetes, but more similar studies are needed on this matter.

Our study demonstrated a higher expression of miR-107 among metformin and insulin users, although significant differences between groups were not found. We expected opposite results. It is known that metformin improves insulin sensitivity, so directly or indirectly should reduce the expression of miR-107, but we got different results. The differences between the results in our study population could be due to long-standing diabetes or long-term insulin treatment. It is also possible that prolonged use of metformin resulted in a reduced effect on insulin sensitivity. However, this is only a hypothesis, and further studies are needed.

However, this study demonstrated a negative correlation between HDL levels and miR-107 levels: Increased expression of miR-107 was related to lower HDL levels, but no such relation was found in diabetic patients. We speculate that the use of statins could interfere with miR-107 expression and diminish the relation of miR-107 and HDL levels in the diabetic population. However, this is the first study that demonstrated a direct relation between miR-107 and HDL levels, and further studies with bigger samples and no use of statins are needed.

Other factors, such as age and gender, were also associated with higher miR-107 levels: Older men had higher miR-107 expression. Interestingly, these factors are also associated with insulin resistance. It has been proven that IR and impaired glucose tolerance are commonly observed phenomena among elderly adults; these are due to reduced mitochondrial function in skeletal muscles and changes in body composition [23]. The association between the male gender and insulin sensitivity has also been noted. According to Karakelides et al., a higher mitochondrial ATP production capacity (MAPR) was noticed in men, whereas women demonstrated higher insulin sensitivity [23]. This dissociation between insulin sensitivity and muscle mitochondrial function, with women showing higher IR and lower muscle MAPR, is still under discussion.

### Study Strengths and Limits

This is the first study that has analyzed the relation of miR-107 expression with clinical conditions, associated with low-grade inflammation, such as diabetes, insulin resistance, obesity, and dyslipidemia. At the same time, we hypothesized that miR-107 could be a messenger molecule that combines diabetes, insulin resistance, obesity, and dyslipidemia. Our indirect results partially support this hypothesis. However, many additional studies are needed to support or exclude this hypothesis.

It would be interesting to evaluate the variability of miR-107 expression in relation to weight changes: Does weight loss have an impact on miR-107 expression? Is the effect due to the reduction of the inflammatory process, or due to the improvement of insulin sensitivity?). Is the expression of miR-107 different in T2DM cases depending on different treatments (oral drugs or insulin therapy)?

However, our study also had some limitations. First and foremost, we consider the sample size. Although some differences were found between groups, the use of a larger sample could demonstrate more reliable results and new links. There is evidence that the HOMA-IR is a useful test for the evaluation of IR, even in patients with T2DM treated with exogenous insulin [24], which is why we analyzed insulin resistance using the HOMA-IR index. However, we are unsure if the HOMA-IR index adequately represents insulin sensitivity in these cases, and we suggest that the measures of IR in the diabetic group should be assessed with caution.

## 5. Conclusions

MiR-107 expression is higher in the case group with diabetic patients than in the group of corresponding nondiabetic control subjects. Higher miR-107 expression is related to dyslipidemia, specifically lower HDL levels, in the general cohort and non-diabetic subjects. Moreover, higher miR-107 expression is related to insulin resistance in the diabetic group. In general, higher miR-107 expression levels are related to a higher BMI, older age, and the male gender.

## Figures and Tables

**Figure 1 biomolecules-12-00558-f001:**
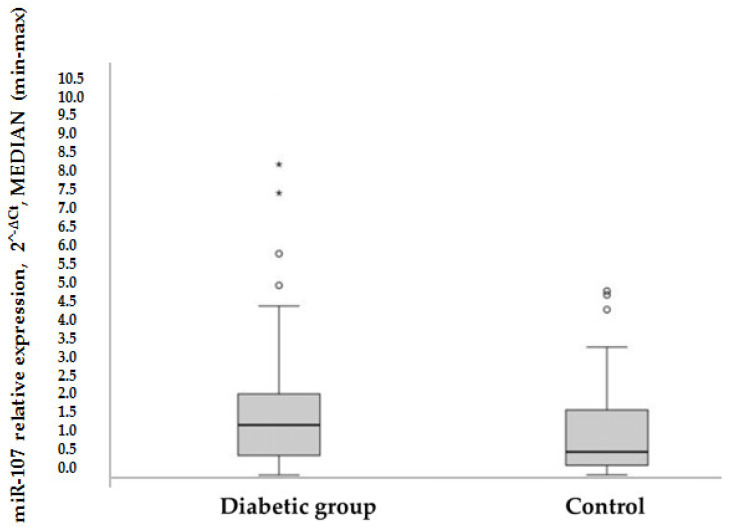
Rectangular chart of patient’s miR-107 expression in T2DM and control groups, *p* = 0.016, according to nonparametric Whitney-Mann test. Data are MEDIAN ± ( max–min); ^o^ and *—displays the distribution of data based on a five number summary (“minimum”, first quartile (Q1), median, third quartile (Q3), and “maximum”). It gives information on outliers and what their values are.

**Figure 2 biomolecules-12-00558-f002:**
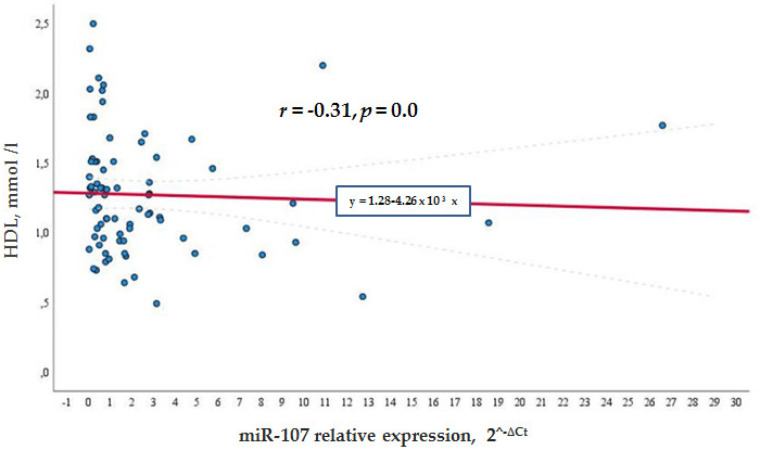
MiR -107 directly and negatively associated with lower HDL levels in the general cohort.

**Figure 3 biomolecules-12-00558-f003:**
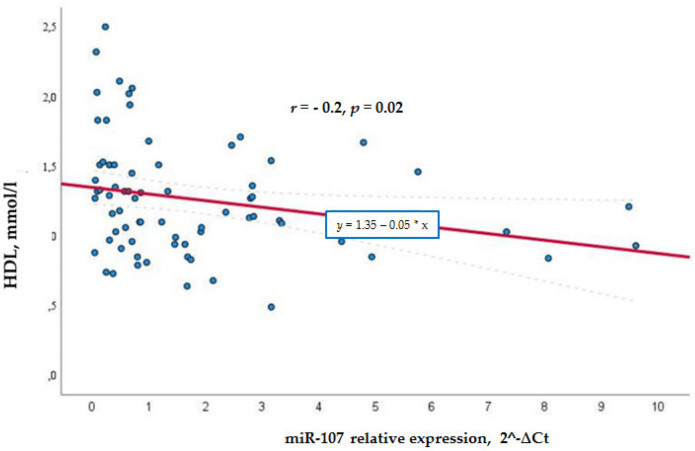
MiR -107 directly and negatively associated with lower HDL levels in the control group.

**Figure 4 biomolecules-12-00558-f004:**
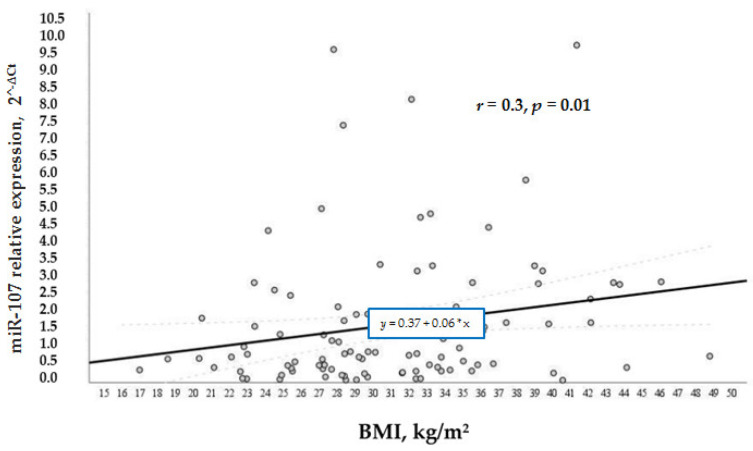
MiR-107 correlation with BMI in the general cohort.

**Figure 5 biomolecules-12-00558-f005:**
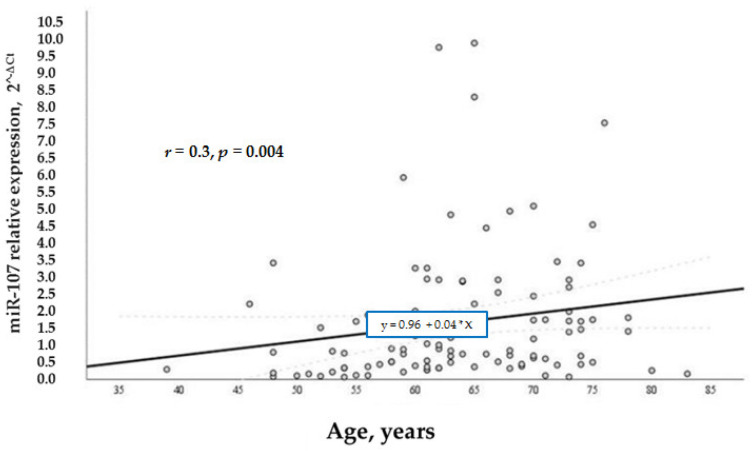
MiR-107 relation with age in the general cohort.

**Figure 6 biomolecules-12-00558-f006:**
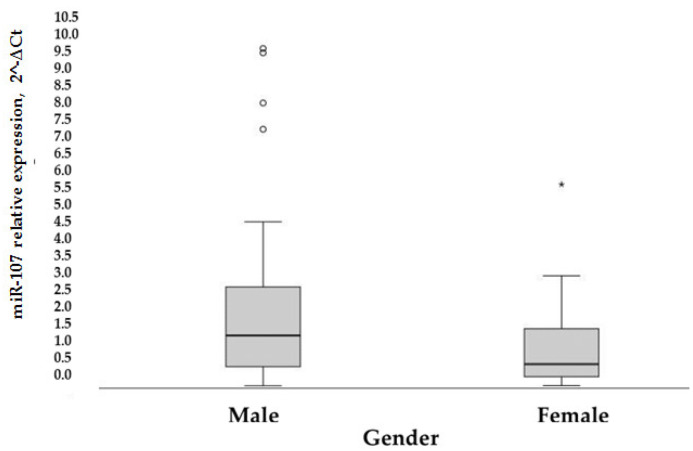
Rectangular chart of patient’s MiR-107 expression, according to gender, *p* = 0.006 according to nonparametric Whitney-Mann test. ^o^ and *—displays the distribution of data based on a five number summary (“minimum”, first quartile (Q1), median, third quartile (Q3), and “maximum”).

**Table 1 biomolecules-12-00558-t001:** General characteristics of study participants. miR-107, HOMA-IR, diabetes duration, HDL, triglycerides are expressed as median (min, max), BMI, HbA1cglycaemia, total cholesterol and LDL as mean ± SD. Abbreviations: NS—nonsignificant, BMI—body mass index, LDL—low density lipids, HDL—high density lipids.

General Characteristics	T2DM Group	Control Group	*p*
Age, years	65 (44–83)	62 (48–80)	NS
Gender: male n (%) female n (%)	24 (45.3%) 29 (54.7%)	25 (46.3%) 29 (53.7%)	NS
BMI, kg/m^2^	34.14 [±5.92]	28.07 [±5.25]	*p* = 0.026
Diabetes duration, years	15 (5–30)	-	-
HbA1c, %	8.23 (±2.14)	5.46 (±0.49)	*p* < 0.001
Glycaemia, mmol/L	-	5.42 (±0.55)	-
Total cholesterol, mmol/L LDL, mmol/L HDL, mmol/L Triglycerides, mmol/L	5.09 (±1.75) 3.20 (±1.16) 1.05 [0.49–2.11] 1.67 [0.57–6.91]	6.23 (±1.38) 4.00 (±1.13) 1.52 [0.96-2.50] 1.02 [0.53-3.57]	*p* = 0.004 *p* = 0.005 *p* < 0.001 *p* < 0.001
Dyslipidaemia rate: Yes, n(%) No, n (%)	40 (76.9%) 12 (23.1%)	11 (36.7%) 19 (63.3%)	*p* < 0.001
HOMA-IR	6.5 [1.2–93.33]	1.9 [0.66–2.9]	*p* < 0.001
miR-107	1.33 [(0.08–9.60]	0.63 [0.04–4.78]	*p* = 0.016
Metformin: yes, n(%) no, n (%)	20 (37.74%) 33 (62.26%)	-	-

## Data Availability

Not applicable.

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
