# Peer review of "Clinical Role of Serum miR107 in Type 2 Diabetes and Related Risk Factors"

_biomolecules, 2022, doi:10.3390/biom12040558_

Round 1

Reviewer 1 Report

  1. The overall language in manuscript needs improvement.
  2. The title of the manuscript is not correct, it is not in accordance with conclusions, and it does not include diabetes
  3. Abstract: line 18 …. whether miR-107 expression in serum is different from

control (with no diabetes) …need clarification

  1. Abstract: Please, add numeric values in Result section
  2. Abstract: There is no conclusion??? Please, add
  3. Line 86. Please, add explanation, are the data from primary, secondary or tertiary care level, in Methodology section…
  4. Methodology: Did you perform 2h OGTT for controls in order to exclude prediabetes and diabetes, please explain
  5. Inclusion criteria: please add explanation why did you include patients who are not o metformin and patients with metformin, having in mind its effect on IR
  6. Please add the number of Approval of Ethics Committee Document
  7. Line 118. Insulin resistance was defined as HOMA-IR above ≥ 2.5., please add reference
  8. Line 166, please add % of pts with T2D using statins
  9. Line 215 … . Thus, weight affects miR-107 expression level….the result differ between diabetics and nondiabetic? Please add explanation
  10. Literature is insufficient, and not up to date, the references are mostly 10 years old

Reviewer 2 Report

“Serum miR-107 a key factor linking dyslipidaemia and obesity”, by Šimonienė et al.

This study aims to examine whether miR-107 expression in serum from diabetic patients is different from control subjects (with no diabetes), whether miR-107 expression correlates with lipid levels, BMI, and whether miR-107 expression is related to insulin resistance.

In general, this study is rather muddled. The English used is not good at all and authors should get help from a native English speaker. Moreover, Figure 3 is missing from the manuscript.

But the most important thing is that the description of the results leaves something to be desired.

In most of the figures, the authors do not compare miR-107 expression from diabetic patients and control subjects (Figure 4, Figure 6 and Figure 7).

Line 193 the authors say "It was found that miR-107 expression directly and positively associated with BMI (r=0.3, p=0.01) (Fig. 4), and age (r=0.3, p=0.004) (Fig. 5). Moreover, overexpression of miR-107 was noticed in men (p=0.006) (Fig. 6). Overall, male gender is related with miR-107 level (p=0.002)". What group is it?

Line 202 "miR-107 correlation with HOMA-IR was evaluated in the groups and in the whole sample". What does it mean?

In conclusion, the authors must review all their results and describe them precisely (comparing those from diabetic patients to those from control subjects).

Round 2

Reviewer 1 Report

  1. Line 27 sentence …Higher miR-107 is related to dyslipidemia lower HDL level- in general cohort ….need clarification
  2. Reference 23. El-Ashmawy N.E, Galway A.M, EL Batanony H. A, Naglaa, F. Khedr N.F, Regulation of MicroRNA 103 and 107 in Obese T2DM Patients Maintained on Metformin. Research square, 2021, PREPRINT (Version 1) should be removed, it is preprint and it is published on the research square (without standard peer review process)

Reviewer 2 Report

The authors made some corrections that I had asked them to make. However, a lot of work remains to be done to improve the text and the presentation of the data.

After re-reading the manuscript, I still think that the level of English is very poor. It is not possible to publish the study as it is. Many turns of phrase do not make sense and need to be corrected.

Moreover, there are still errors in the text: for example, line 179, the statistical calculation is meaningless (p=0.000).

The authors say that they do not see a difference in miR-107 expression between control subjects and diabetic patients (probably due to a too small population). However, they still have to specify this in the text. It is important to compare what happens between the two groups studied.

Line 196, figure 2, the authors wrote "Mir-107 directly and negatively associated with lower HDL level in general". What does "in general" mean? The authors need to be much more descriptive in the description of the figures. "In general," is not really a scientific term to use in an article to describe results.

Overall, the figure legends are not precise enough and do not describe the results sufficiently. Thus, they need to be revised.
